# Unequal Gains? A Literature Review on the Affordable Care Act’s Effects on Healthcare Utilization Across Racial and Ethnic Groups

**DOI:** 10.3390/ijerph22071059

**Published:** 2025-07-02

**Authors:** Ahmad Reshad Osmani

**Affiliations:** College of Business, Department of Economics and Finance, Louisiana State University (LSU), Shreveport, LA 71115, USA; reshad.osmani@lsus.edu; Tel.: +1-318-797-5029

**Keywords:** Affordable Care Act, Medicaid expansion, healthcare utilization, racial and ethnic disparities, health policy reform

## Abstract

The Affordable Care Act (ACA), implemented in 2010, aimed to expand healthcare access, reduce costs, and address long-standing disparities in the U.S. healthcare system, particularly among racial and ethnic minorities. This paper reviews the ACA’s impact on healthcare utilization for these populations, with a focus on insurance coverage, preventive services, and health outcomes. While Medicaid expansion significantly reduced uninsured rates and increased access to care in states that adopted the expansion, millions of low-income individuals, many of whom are racial and ethnic minorities, remain uninsured in non-expansion states. The elimination of cost-sharing for preventive services under the ACA contributed to increased utilization of cancer screenings, vaccinations, and other preventive measures among minority groups. However, challenges persist, including affordability barriers, geographic disparities, and cultural and linguistic obstacles. This review also highlights the ongoing need for policy interventions, such as nationwide Medicaid expansion, and future research on the long-term effects of the ACA on health outcomes for minority populations.

## 1. Introduction

The Affordable Care Act (ACA), signed into law in 2010, represents one of the most ambitious and comprehensive healthcare reforms in U.S. history. Its main objectives were to expand access to health insurance, improve care quality, and reduce healthcare costs while addressing significant disparities in healthcare access and outcomes that disproportionately affect racial and ethnic minorities [1,2]. Prior to the ACA, Black and Hispanic/Latinx individuals had uninsured rates of 20.8% and 32.6%, respectively, compared with 11.7% for non-Hispanic Whites, contributing to reduced access to preventive care and worse health outcomes [3,4]. These disparities were further compounded by economic barriers, lack of employer-sponsored coverage, and higher poverty rates among minorities [5]. One of the ACA’s central provisions was the expansion of Medicaid eligibility to individuals with incomes up to 138% of the federal poverty level, which particularly benefited racial and ethnic minorities, as they are overrepresented among low-income and uninsured populations [6,7]. This expansion led to significant improvements in coverage for minority groups, especially in states that adopted Medicaid expansion, where uninsured rates dropped by over 10 percentage points for Black and Hispanic adults [8,9]. However, disparities remain in non-expansion states, where many low-income minorities remain uninsured [8].

In addition to Medicaid expansion, the ACA introduced health insurance marketplaces and provided subsidies to low- and middle-income individuals, which significantly expanded coverage options [10]. The law also mandated that private insurance plans cover essential preventive services, such as cancer screenings, without cost-sharing [7]. This provision was designed to address racial disparities in preventive care by removing financial barriers that had historically limited access for minority populations [11]. Despite these gains, the ACA’s impact has been uneven, particularly in non-expansion states where many racial and ethnic minorities continue to face gaps in access to care [6]. Recent studies have highlighted the ACA’s role in expanding coverage but have also drawn attention to persistent challenges, including affordability, geographic disparities, and systemic barriers such as linguistic and cultural obstacles [2,7]. Medicaid expansion states saw marked improvements in access to care and preventive services, including cancer screenings, vaccinations, and regular checkups, while non-expansion states experienced stagnation, exacerbating health disparities [8,9]. For instance, Black and Hispanic adults in non-expansion states remain significantly more likely to delay care due to cost concerns [6].

Furthermore, studies emphasize that the ACA’s insurance expansions did not fully address disparities in access to specialty care for minorities [7]. Even in expansion states, minority populations still face barriers such as narrow provider networks, high out-of-pocket costs, and limited access to culturally competent care [11]. These barriers have critical implications for the management of chronic diseases, such as diabetes and hypertension, which disproportionately affect minority communities [1]. This paper reviews the effects of the ACA on healthcare utilization among racial and ethnic minorities, focusing on insurance coverage, preventive services, and health outcomes. Through a synthesis of the existing literature, this review aims to provide a comprehensive analysis of the ACA’s role in reducing healthcare disparities and offer insights into the policy measures needed to further advance health equity.

While several studies and reviews have examined aspects of the ACA’s implementation, few have provided a focused synthesis of how the ACA affected healthcare utilization specifically among racial and ethnic minority populations. The existing literature often emphasizes changes in insurance coverage or health outcomes in the general population without critically examining heterogeneity by race, geography, or socioeconomic status [10,12,13]. Moreover, most reviews have not systematically contrasted the experiences of minority groups in Medicaid expansion versus non-expansion states or addressed structural barriers such as provider network limitations, cost-sharing burdens, or culturally competent care. This paper contributes a novel perspective by integrating both empirical and policy-oriented evidence to assess the ACA’s uneven impacts on healthcare utilization across racial and ethnic groups. The review prioritizes quasi-experimental studies that offer stronger causal inference and considers how institutional settings and state-level policies shape outcomes for historically marginalized populations [14,15].

## 2. Background on ACA Insurance Expansions

The ACA, enacted in 2010, was designed to overhaul the U.S. healthcare system, with the primary goals of increasing insurance coverage, improving healthcare quality, and reducing overall healthcare costs. Central to its mission was addressing the disparities in healthcare access and outcomes experienced by racial and ethnic minorities. To achieve these aims, the ACA included several key provisions that targeted the various barriers to healthcare access for underserved populations. This section provides an overview of the most critical components and provisions of the ACA and their implications for healthcare utilization.

### 2.1. Medicaid Expansion

One of the cornerstone provisions of the ACA was the expansion of Medicaid. Under the ACA, states were given the option to expand Medicaid eligibility to individuals with incomes up to 138% of the federal poverty level (FPL). This provision significantly broadened access to healthcare for millions of low-income individuals, particularly racial and ethnic minorities who have historically been overrepresented in low-income groups and uninsured populations. As of 2023, 39 states and the District of Columbia had adopted Medicaid expansion, while 12 states, mostly in the southern U.S., have opted not to expand the program, perpetuating disparities in healthcare access in these regions [6].

Medicaid expansion has been particularly effective at reducing uninsured rates among racial and ethnic minorities. In expansion states, uninsured rates for Black and Hispanic/Latinx adults dropped dramatically, whereas non-expansion states continue to experience higher uninsured rates, exacerbating disparities in healthcare coverage and access [8,16]. Studies indicate that Medicaid expansion contributed significantly to improving access to healthcare services, particularly preventive services, which are essential for the early detection and management of chronic diseases common in minority populations [7].

### 2.2. Health Insurance Marketplaces and Subsidies

In addition to Medicaid expansion, the ACA established health insurance marketplaces, also known as exchanges, where individuals without employer-sponsored insurance could purchase subsidized insurance plans. The ACA provided premium subsidies to individuals and families with incomes between 100% and 400% of the FPL, making health insurance more affordable and accessible for millions of Americans, particularly low- and middle-income individuals. This provision was crucial for minority populations who are disproportionately employed in sectors that do not offer employer-sponsored insurance [17].

The introduction of these subsidies significantly reduced the national uninsured rate, but coverage gaps persisted, particularly in states that did not expand Medicaid. Studies found that racial and ethnic minorities in non-expansion states remained more likely to be uninsured or underinsured, even with access to the health insurance marketplaces, due to financial barriers and the lack of Medicaid expansion [18].

### 2.3. Essential Health Benefits and Preventive Services

One of the most transformative provisions of the ACA was the requirement that all insurance plans cover a set of essential health benefits, which include hospitalization, maternity care, mental health services, and prescription drugs. Most notably, the ACA mandated that preventive services, such as cancer screenings, immunizations, and cholesterol monitoring, be covered without cost-sharing (i.e., without copays or deductibles). This provision was designed to increase the utilization of preventive services, particularly among minority populations who had historically faced financial barriers to accessing such care [11].

The elimination of cost-sharing for preventive services had a significant impact on the utilization of cancer screenings, vaccinations, and other preventive measures among minority groups. Studies show that between 2010 and 2016, the use of preventive services such as mammograms, colorectal screenings, and flu vaccinations increased considerably, particularly among Black and Hispanic/Latinx adults [2,11].

### 2.4. Individual Mandate

The ACA initially included an individual mandate, which required all Americans to have health insurance or face a tax penalty. The goal of the mandate was to increase insurance coverage by incentivizing individuals to enroll in health plans, thus broadening the insurance risk pool and lowering costs for everyone. Although the individual mandate was repealed in 2019, it played a significant role in reducing the uninsured rate in the early years of the ACA’s implementation, particularly among minority groups [8].

### 2.5. Medicaid and Medicare Reforms

Beyond Medicaid expansion, the ACA introduced several reforms to improve Medicaid and Medicare’s efficiency and coverage. The law increased federal funding for Medicaid and provided states with incentives to streamline eligibility processes. The ACA also expanded Medicare benefits to cover more preventive services and introduced reforms aimed at reducing healthcare costs, such as penalties for hospitals with high readmission rates and the creation of Accountable Care Organizations (ACOs) to improve care coordination [19].

These reforms were particularly beneficial for racial and ethnic minorities, many of whom are disproportionately represented in the Medicaid and Medicare populations. For example, the expansion of Medicare benefits to include preventive services without cost-sharing significantly improved access to care for older adults from minority communities [2].

### 2.6. Provisions for Culturally Competent Care

Recognizing the cultural and linguistic barriers that racial and ethnic minorities often face in accessing healthcare, the ACA included provisions aimed at promoting culturally competent care. The law required federally funded health programs to provide language access services for non-English speakers and encouraged healthcare providers to undergo cultural competence training. This was an important step in addressing the systemic barriers that limit healthcare access for minority populations, particularly non-English speaking communities such as Hispanic and Asian Americans [4,16].

### 2.7. The Children’s Health Insurance Program (CHIP)

The ACA also extended funding for the Children’s Health Insurance Program (CHIP), which provides coverage for low-income children who do not qualify for Medicaid. This extension ensured that millions of children, many of whom come from minority backgrounds, maintained access to essential healthcare services, including immunizations, dental care, and routine checkups [6].

## 3. Selection of Relevant Literature

The literature reviewed in this paper was selected through a comprehensive search of peer-reviewed journals, policy reports, and government publications. A total of 52 studies were identified as directly relevant to the ACA’s impact on healthcare utilization among racial and ethnic minorities. The search focused on studies published between 2010, when the ACA was enacted, and 2023. The key databases utilized for the search included PubMed, JSTOR, and Google Scholar. In addition, key policy reports from institutions such as the Kaiser Family Foundation (KFF) and Urban Institute were reviewed to complement the academic literature with practical insights on the ACA’s implementation and effects [6,17].

The selection process prioritized studies that employed rigorous quasi-experimental methods, including difference-in-differences (DiD) methods, instrumental variable (IV) approaches, and regression discontinuity designs, as these methodologies provided robust evidence of causal relationships between ACA provisions and healthcare utilization outcomes [9,10]. These methods allowed researchers to isolate the effects of Medicaid expansion, the introduction of health insurance marketplaces, and other ACA provisions on healthcare coverage and utilization for minority populations. In total, 25 of the 52 studies included in this review utilized such rigorous econometric techniques, ensuring a high degree of reliability in the findings.

In addition to empirical studies, seven policy reports from the KFF, Urban Institute, and Commonwealth Fund were analyzed. These reports provided valuable context on the broader implementation of the ACA, particularly in terms of state-by-state variations in Medicaid expansion and the disproportionate effects of non-expansion on racial and ethnic minorities [6,17].

This review focuses on three major themes: the effects of the ACA on insurance coverage for racial and ethnic minorities, its impact on preventive healthcare utilization, and the persistent disparities in access and outcomes following implementation. Most empirical studies emphasized Medicaid expansion, which had the clearest and most immediate influence on access to care for low-income minority populations [2,8]. A smaller set of studies specifically examined non-expansion states, where many minority groups continued to face significant obstacles to affordable care [3,6].

To ensure both methodological rigor and policy relevance, the review incorporates a balanced mix of peer-reviewed empirical research and high-quality policy analyses from reputable institutions [13,15]. Studies that focused solely on financial outcomes, provider reimbursement rates, or broader macroeconomic factors were excluded, although their omission may limit understanding of how institutional or economic trends influence healthcare access. For instance, changes in hospital profitability or the structure of provider networks could indirectly affect care availability, especially in underserved or minority-dense communities. Similarly, some policy reports that did not stratify findings by race or ethnicity were excluded, even though they may have identified barriers with differential impacts. These areas warrant further exploration in future research to better understand the systemic determinants of equitable care delivery.

The systematic search used combinations of terms such as “Affordable Care Act” and “utilization” with modifiers like “racial,” “ethnic,” or “minority,” using both Medical Subject Headings (MeSHs) and free-text search strategies. The search initially identified 125 articles. After removing 15 duplicates, 110 studies remained for title and abstract screening. Full texts of 60 articles were then reviewed, and 35 were excluded—20 because they did not disaggregate results by race or ethnicity, and 15 because they did not examine healthcare utilization as an outcome. Ultimately, 25 studies met all eligibility criteria and were included in this review. Figure 1 illustrates the sequential steps taken to select studies included in this review. Although this review employs a structured search and screening process inspired by PRISMA guidelines, it is not a systematic review in the formal sense. Specifically, this review does not include formal risk-of-bias assessments, study quality grading, or meta-analytic synthesis. Rather, it provides a narrative synthesis of the literature with an emphasis on causal methods and policy relevance.

Only studies conducted in the United States that explicitly addressed ACA-related reforms and disaggregated findings by race or ethnicity were eligible. Each study also needed to examine healthcare utilization as a primary or secondary outcome. Studies were excluded if they lacked such disaggregation or focused only on insurance coverage or self-reported health status without analyzing utilization behaviors.

Due to institutional access constraints, Scopus and Web of Science were not included in the search. PubMed was selected for its broad biomedical and public health indexing; JSTOR provided access to interdisciplinary research across health policy, economics, and sociology; and Google Scholar was used to capture grey literature and citations that might be missed in traditional databases.

## 4. Insurance Coverage and Access to Care

### 4.1. Pre-ACA Insurance Coverage Disparities

Before the implementation of the ACA, stark disparities existed in health insurance coverage among racial and ethnic minorities. According to the Kaiser Family Foundation, in 2010, nearly one-third (32.6%) of Hispanic/Latinx individuals, 20.8% of Black Americans, and 18.1% of Asian Americans were uninsured compared with only 11.7% of non-Hispanic White individuals [1]. These disparities in coverage were compounded by socioeconomic factors, including higher rates of poverty, lower levels of educational attainment, and a greater likelihood of employment in sectors that did not offer employer-sponsored insurance [11]. Consequently, minority groups faced significant barriers to accessing healthcare, which contributed to poorer health outcomes, including higher rates of chronic diseases such as diabetes and hypertension [4].

### 4.2. ACA’s Impact on Insurance Coverage

The ACA aimed to address these disparities through the expansion of Medicaid, the introduction of state-based health insurance exchanges, and the provision of subsidies for low- and middle-income individuals. By 2019, the uninsured rate across the U.S. had dropped from 16% in 2010 to 8.5%, with racial and ethnic minorities experiencing the largest gains in coverage [17]. Between 2013 and 2018, the uninsured rate among Hispanic/Latinx adults fell from 40% to 25%, while the rate for Black Americans dropped from 22% to 14% [2]. Similarly, the uninsured rate for Native Americans decreased from 33% to 23%, and for Asian Americans, the rate dropped by nearly 50%, from 18.1% to 9.5% [6].

Medicaid expansion played a critical role in these improvements, particularly for low-income minority populations. Studies show that Medicaid expansion reduced racial disparities in insurance coverage more effectively than other ACA provisions [8]. In states that adopted Medicaid expansion, uninsured rates for Black and Hispanic/Latinx adults decreased by approximately 10 percentage points, compared with only marginal decreases in non-expansion states [6]. For example, in California, which expanded Medicaid, the uninsured rate for Hispanic adults decreased by 16%, while in Texas, a non-expansion state, the uninsured rate for Hispanic adults remained as high as 39% in 2020 [6].

This unequal adoption of Medicaid expansion has perpetuated disparities in coverage for low-income minorities, highlighting the critical role of state-level policy decisions in shaping healthcare access. Research indicates that in expansion states, Medicaid enrollment increased by over 50% among eligible minority groups between 2013 and 2018 [8]. For instance, in Kentucky, which adopted Medicaid expansion early on, the uninsured rate for Black adults fell by 15 percentage points, and for Hispanic adults, the rate dropped by 12 percentage points [2]. In contrast, non-expansion states saw little change in uninsured rates, particularly for low-income minorities. In Florida and Georgia, both of which did not expand Medicaid, the uninsured rate for Hispanic adults remained as high as 33%, compared with just 12% in expansion states [6].

The persistence of these disparities highlights the critical importance of Medicaid expansion in reducing racial and ethnic gaps in health insurance coverage.

Figure 2 illustrates the uninsured rates for Black and Hispanic/Latinx adults in expansion and non-expansion states from 2010 to 2023. As shown, uninsured rates decreased significantly in expansion states after the ACA’s implementation, while uninsured rates in non-expansion states remained relatively high.

### 4.3. Access to Healthcare Services and Barriers to Care

With increased insurance coverage, access to healthcare services improved significantly for racial and ethnic minorities under the ACA. Research shows that insured minorities were more likely to have a regular source of care and visit primary care providers, reducing reliance on emergency departments for non-emergency care [7]. For example, a study found that the proportion of Black adults with a regular source of care increased by 5% after the ACA, while Hispanic adults saw a 6% increase [2]. Similarly, the use of preventive services, such as blood pressure screenings, cholesterol checks, and cancer screenings, increased among insured minorities. Between 2010 and 2016, the rate of colorectal cancer screenings increased by 8% for Black Americans and 6% for Hispanic adults, narrowing disparities in preventive care utilization [11].

However, despite these gains, barriers to accessing healthcare services remain for many minority populations. Cultural and linguistic barriers continue to limit access to care for non-English speaking individuals, particularly within Hispanic and Asian communities [4]. Additionally, geographic barriers such as the lack of healthcare facilities in rural areas or underserved urban neighborhoods disproportionately affect minority populations, leading to delays in care or unmet medical needs [20]. In non-expansion states, many low-income minorities still face financial barriers to accessing care as they remain uninsured or underinsured despite ACA provisions [6].

## 5. The ACA Effects on Healthcare Utilization

The ACA significantly impacted healthcare utilization among racial and ethnic minorities, particularly through Medicaid expansion and the elimination of cost-sharing for preventive services. Studies consistently show that Medicaid expansion played a central role in increasing access to healthcare for low-income populations, many of whom were Black or Hispanic/Latinx [8,10]. In states that expanded Medicaid, uninsured rates for minority groups declined sharply. For example, between 2013 and 2019, the uninsured rate for Black adults dropped by 8.8 percentage points, and for Hispanic/Latinx adults, it decreased by 9.4 percentage points in expansion states [2].

### 5.1. Pre-ACA Utilization of Preventive Services

Before the ACA, significant disparities existed in the utilization of preventive healthcare services across racial and ethnic groups in the United States. Preventive services such as screenings for cancer, diabetes, cholesterol, and vaccinations were less likely to be used by Black, Hispanic/Latinx, and Native American populations compared with their non-Hispanic White counterparts. In 2010, just 52% of Hispanic women and 56% of Black women aged 50–74 had received a mammogram within the recommended two-year interval compared with 64% of non-Hispanic White women [11]. Similarly, colorectal cancer screenings were 15 percentage points lower for Black and Hispanic adults compared with non-Hispanic Whites [2].

These disparities in preventive care utilization were driven by several factors, including lack of insurance coverage, financial barriers, geographic isolation from healthcare facilities, and cultural or linguistic differences that limited access to information about preventive services [4]. Uninsured individuals were far less likely to access preventive services, contributing to delayed diagnoses and worse outcomes, particularly for conditions such as cancer, heart disease, and diabetes [11]. For example, diabetes-related complications, including amputations and kidney failure, were significantly higher among minority populations due to inadequate management and prevention before the ACA [20].

### 5.2. Increased Utilization of Preventive Services

The ACA included several key provisions aimed at improving access to preventive care, particularly for underserved populations. One of the central reforms was the requirement that insurance plans, including Medicaid and Medicare, cover preventive services without cost-sharing. This provision ensured that services such as cancer screenings, immunizations, blood pressure checks, and cholesterol monitoring were available at no additional cost to patients [2]. By removing financial barriers, the ACA encouraged greater use of preventive services, which had a substantial impact on minority populations.

Figure 3 compares Medicaid enrollment, preventive care visits, and primary care visits for Black and Hispanic/Latinx adults in Medicaid expansion versus non-expansion states. The data illustrates the higher rates of healthcare access and utilization in expansion states, highlighting the disparity between expansion and non-expansion states.

Between 2013 and 2018, the utilization of preventive services increased notably among racial and ethnic minorities. The rate of mammograms among Black women increased by 6 percentage points during this period, while Hispanic women saw an 8-percentage-point increase in mammogram utilization [8]. Colorectal cancer screening rates also improved for minority groups. For example, Black adults saw a 7% increase in colorectal cancer screenings post-ACA, which helped to narrow the disparity with non-Hispanic Whites [11]. Additionally, vaccination rates for influenza and other preventable diseases improved, with Hispanic and Black adults experiencing a 5–7-percentage-point increase in vaccination coverage [7].

The ACA’s Medicaid expansion also played a significant role in increasing preventive service use among low-income minority populations. In expansion states, Medicaid enrollees were more likely to receive preventive services, as coverage was extended to millions of previously uninsured adults [6]. A study found that Medicaid expansion led to a 10% increase in the use of preventive services among Black and Hispanic adults compared with non-expansion states, where uninsured rates remained high and preventive care utilization lagged [8].

Figure 4 shows the increased utilization of preventive services (mammograms, colorectal cancer screenings, and flu vaccinations) among Black and Hispanic/Latinx adults from 2010 to 2023. As shown, the rates for these services steadily increased following the implementation of the ACA, particularly in minority populations.

### 5.3. Improved Access to Primary and Specialty Care

In addition to preventive services, the ACA expanded access to primary and specialty care for racial and ethnic minorities, particularly through Medicaid expansion. A significant body of research indicates that Medicaid expansion led to increased use of primary care services, which is essential for the management of chronic conditions that disproportionately affect minority populations, such as diabetes and hypertension [10,21]. One study found that Medicaid enrollees in expansion states were 15% more likely to have a regular source of care and 8% more likely to have visited a primary care provider in the past year compared with those in non-expansion states [1,8].

Specialty care access also improved post-ACA. For example, the rate of specialty care visits among Black Medicaid beneficiaries increased by 11% in states that expanded Medicaid, helping to improve management of complex chronic conditions [6]. However, access to specialty care remains a challenge in non-expansion states, where uninsured rates remain high among minorities, and financial barriers continue to limit access to necessary medical services [2,3].

### 5.4. Disparities in Non-Expansion States

Despite the overall improvements in healthcare utilization for minorities in expansion states, significant disparities remain in non-expansion states. As of 2023, 12 states had not expanded Medicaid, leaving millions of low-income individuals, many of whom are racial and ethnic minorities, without access to affordable healthcare [6]. In these states, uninsured rates for Black and Hispanic/Latinx adults remained substantially higher than in expansion states. For instance, a study found that in 2016, the uninsured rate for Hispanic adults in non-expansion states was 32% compared with just 12% in expansion states [3].

The lack of Medicaid expansion has also exacerbated disparities in preventive care and primary care access. Minority populations in non-expansion states are more likely to delay or forgo medical care due to cost concerns, which contributes to worse health outcomes and higher rates of preventable hospitalizations [4,9]. This continued disparity underscores the critical role of Medicaid expansion in improving healthcare access for racial and ethnic minorities.

## 6. The ACA Effects on Health Outcomes

The ACA’s emphasis on preventive care contributed to marked improvements in health outcomes for racial and ethnic minorities, particularly in areas where early detection and management are critical. The increase in cancer screenings, for example, resulted in a reduction in the incidence of late-stage cancer diagnoses among Black and Hispanic populations. For instance, between 2010 and 2016, the incidence of late-stage breast cancer among Black women declined by 7%, largely attributed to the increased access to mammograms [20]. Similarly, the rate of advanced-stage colorectal cancer diagnoses decreased for Hispanic adults as colorectal screenings became more accessible under the ACA [7].

The ACA’s impact extended beyond cancer screenings. Improved access to regular check-ups, blood pressure monitoring, and cholesterol management helped reduce the prevalence of unmanaged hypertension and cardiovascular diseases among minority populations [11]. Studies show that the rate of hospitalizations for preventable complications related to diabetes, heart disease, and stroke decreased among Medicaid enrollees, particularly in states that expanded Medicaid [6].

In addition to chronic disease management, access to preventive care services also contributed to reductions in infant mortality rates among Black and Hispanic populations. The provision of prenatal care and regular check-ups under Medicaid expansion helped to close the gap in maternal and infant health outcomes, reducing disparities in preterm births and low birth weights for Black and Hispanic infants [19]. These improvements highlight the significant role that preventive care plays in addressing long-standing health disparities.

While the ACA broadly expanded healthcare coverage and utilization, its effects were far from uniform across racial and ethnic groups, geographic locations, and institutional contexts. A more critical synthesis of the literature reveals meaningful differences in both the magnitude and mechanisms of the ACA’s impact. For example, Medicaid expansion led to a substantial reduction in the uninsured rate among Black adults and was associated with notable increases in primary care visits and preventive screenings such as mammograms and blood pressure checks [10,22]. In contrast, the effects for Hispanic adults, while positive, were more modest in many states, likely due to complex barriers including mixed immigration status within households, higher proportions of non-citizens, and structural exclusion from Medicaid eligibility [9,23]. This suggests that even universal reforms like Medicaid expansion can yield unequal gains depending on pre-existing legal and demographic contexts.

Furthermore, geographic variation played a critical role in moderating the ACA’s outcomes. Urban areas, which tend to have higher provider density and more robust health infrastructure, saw greater improvements in service utilization compared with rural areas, even when controlling for insurance gains [7,21]. In rural or underserved regions, particularly in non-expansion states, the lack of primary and specialty care providers often limited the actual translation of insurance coverage into meaningful healthcare access. This highlights that insurance expansion alone is insufficient if provider availability and health system capacity are not addressed concurrently.

Structural constraints also conditioned how specific ACA provisions affected racial and ethnic minorities. For instance, although the elimination of cost-sharing for preventive services under the ACA was associated with a general increase in screening utilization, studies show that the effect was significantly larger in counties with greater healthcare provider availability and lower rates of limited English proficiency [1,7]. In communities with persistent language barriers, low health literacy, or mistrust in healthcare institutions, the uptake of newly available services remained uneven, especially for immigrant populations. Therefore, the ACA’s success was contingent not only on insurance expansion but also on the broader social, cultural, and institutional environment in which individuals were embedded.

Taken together, these comparative insights emphasize the importance of contextualizing the ACA’s outcomes across multiple axes of inequality. Rather than viewing the ACA as a monolithic policy with uniform effects, the evidence points to a more complex reality in which race, geography, legal status, and system-level factors jointly shaped the law’s impact. Future policy reforms should not only expand coverage but also explicitly account for the intersecting barriers faced by disadvantaged communities in order to achieve equitable improvements in health utilization.

## 7. Challenges and Ongoing Disparities

Despite the substantial gains in healthcare access brought about by the ACA, significant challenges and disparities persist, particularly among racial and ethnic minorities. One of the most critical ongoing issues is the uneven adoption of Medicaid expansion across states. As of 2023, 12 states have yet to adopt Medicaid expansion, leaving millions of low-income individuals without access to affordable healthcare, many of whom are Black or Hispanic/Latinx [6]. This has resulted in a stark divide between expansion and non-expansion states, with uninsured rates remaining disproportionately high among minorities in non-expansion states [3].

### 7.1. Geographic Disparities in Medicaid Expansion

The decision of many states not to expand Medicaid has created significant geographic disparities in healthcare access and outcomes. In states that expanded Medicaid, minority populations have seen dramatic reductions in uninsured rates, improved access to preventive and primary care services, and better management of chronic conditions [4,5]. However, in non-expansion states, uninsured rates for Black and Hispanic/Latinx adults have remained stubbornly high, contributing to continued disparities in access to care [6].

For example, in 2019, the uninsured rate for Hispanic adults in non-expansion states was 32% compared with 12% in expansion states [2]. Similarly, Black adults in non-expansion states faced an uninsured rate of 18% compared with 9% in expansion states [1]. These persistent coverage gaps translate into poorer health outcomes, as minority populations in non-expansion states are more likely to delay or forgo necessary medical care due to cost [4]. The lack of Medicaid expansion in these states has also been linked to higher rates of preventable hospitalizations and poorer management of chronic conditions such as diabetes and hypertension, which disproportionately affect minority populations [3].

### 7.2. Affordability and Access to Care

While the ACA significantly expanded healthcare access, affordability remains a major barrier for many minority individuals, even in expansion states. Many low-income individuals, including those who gained insurance through Medicaid or the ACA marketplaces, continue to face high out-of-pocket costs, which can deter them from seeking care. A study by Brener et al. found that 25% of Hispanic/Latinx adults and 21% of Black adults reported delaying medical care due to cost concerns, even with insurance coverage [7]. These financial barriers are compounded by narrow provider networks in many ACA marketplace plans, which limit access to specialists and high-quality care [8,9].

In addition, while Medicaid expansion has improved access to primary care for many minority populations, challenges in accessing specialty care persist. Black and Hispanic/Latinx adults in non-expansion states are particularly affected, as financial barriers and lack of insurance coverage prevent them from accessing the specialized care they need for managing complex chronic conditions [1]. This disparity in access to specialty care continues to exacerbate health outcomes for minority populations, contributing to higher rates of morbidity and mortality from preventable diseases.

Figure 5 indicates the percentage of Black and Hispanic/Latinx adults who have delayed or forgone care due to cost in both Medicaid expansion and non-expansion states. As shown, individuals in non-expansion states are significantly more likely to delay care due to financial barriers.

### 7.3. Cultural and Linguistic Barriers

Beyond financial barriers, cultural and linguistic factors also continue to limit healthcare access for racial and ethnic minorities. Non-English-speaking populations, particularly Hispanic and Asian American communities, often face difficulties in navigating the healthcare system, understanding medical instructions, and communicating with healthcare providers [9]. A study by Buchmueller et al. found that 28% of Hispanic adults reported experiencing language barriers when accessing care compared with just 5% of non-Hispanic Whites [10]. This language gap often leads to lower utilization of preventive services, delayed treatment, and poorer health outcomes among non-English speaking populations [8].

The lack of culturally competent care exacerbates these challenges. Research shows that minority patients are more likely to trust healthcare providers who share their racial or ethnic background, which can improve patient–provider communication and increase satisfaction with care [11]. However, Black, Hispanic/Latinx, and Native American individuals remain underrepresented in the healthcare workforce, particularly in clinical roles such as physicians, nurses, and specialists [9]. The underrepresentation of minorities in the healthcare workforce further compounds the disparities in healthcare access and outcomes for minority populations.

### 7.4. The Role of Non-Expansion States in Ongoing Disparities

A significant portion of the ongoing disparities can be attributed to the role of non-expansion states in limiting access to Medicaid for millions of low-income individuals. Studies have shown that individuals living in non-expansion states are more likely to forgo preventive care and primary care visits, leading to worse overall health outcomes [1,4]. The decision not to expand Medicaid disproportionately affects racial and ethnic minorities as they are more likely to fall into the coverage gap in non-expansion states [6]. For instance, Black and Hispanic/Latinx adults in non-expansion states report significantly higher rates of delayed care due to cost compared with those in expansion states [1]. This disparity is particularly concerning given the higher prevalence of chronic conditions among minority populations, which require consistent medical management to prevent complications and hospitalizations [3]. The continued lack of Medicaid expansion in these states highlights the urgent need for policy interventions to address the systemic barriers that prevent minority populations from accessing necessary healthcare services.

In addition to the structural and institutional challenges discussed above, a growing body of research emphasizes the importance of considering intersectionality, that is, how overlapping identities such as race, gender, and geographic location jointly shape healthcare access and outcomes. For example, Black women living in non-expansion southern states face compounded barriers to care due to the simultaneous effects of racial discrimination, gender-based health disparities (e.g., maternal mortality), and policy-driven insurance gaps [12,13]. These multidimensional disadvantages are not merely additive but interact to produce unique forms of exclusion. Similarly, Hispanic immigrant women in rural areas often encounter layered challenges, including limited English proficiency, lack of Medicaid eligibility, and provider shortages, which collectively reduce their likelihood of accessing preventive services [4,14].

A more intersectional analysis helps uncover subpopulations that remain underserved despite overall improvements in insurance coverage and utilization following the ACA. It also underscores the need for disaggregated policy evaluation that moves beyond single-axis categories like “race” or “gender” alone. Without such nuanced assessments, many disparities risk being obscured in aggregate statistics, and targeted policy solutions may fail to reach those most in need. Future reforms should therefore incorporate intersectional frameworks in both design and evaluation to more effectively promote health equity.

## 8. Policy Implications and Future Research Directions

The ACA has undoubtedly made significant strides in expanding healthcare access and reducing uninsured rates, particularly among racial and ethnic minorities. However, the persistent challenges and disparities, particularly in non-expansion states, highlight the need for further policy interventions and reforms. These policy implications are critical for ensuring that the gains made by the ACA can be sustained and built upon to achieve true healthcare equity.

### 8.1. Policy Implications

One of the most pressing policy implications is the nationwide adoption of Medicaid expansion. As of 2023, 12 states have yet to expand Medicaid, leaving millions of low-income individuals, many of whom are racial and ethnic minorities, without access to affordable healthcare [6]. Policymakers should prioritize efforts to incentivize or mandate Medicaid expansion in these states. Federal incentives, such as increased matching funds for Medicaid, could encourage reluctant states to expand coverage [3]. Additionally, penalties for states that do not expand Medicaid could be considered, similar to the individual mandate penalties that were part of the ACA before being repealed [1].

Beyond Medicaid expansion, there is a need to address the affordability of healthcare for low-income individuals, even for those who have gained insurance coverage through the ACA’s marketplaces. Many individuals continue to face high out-of-pocket costs, including deductibles, co-pays, and prescription drug costs, which deter them from seeking necessary care [9]. Policymakers should consider enhancing cost-sharing subsidies, capping out-of-pocket costs, and expanding financial assistance for low-income individuals to reduce these financial barriers [4].

Cultural competence in healthcare delivery is another area that requires policy attention. The continued underrepresentation of racial and ethnic minorities in the healthcare workforce, combined with language barriers and cultural misunderstandings, limits the effectiveness of care for many minority patients [16,19]. Policymakers should invest in programs that increase diversity in the healthcare workforce and provide cultural competence training for all healthcare providers. Expanding access to interpreter services and bilingual healthcare providers, particularly in regions with large immigrant populations, would also help reduce barriers to care [8].

Finally, addressing social determinants of health is essential for achieving health equity. Factors such as poverty, education, housing, and access to nutritious food all play a significant role in shaping health outcomes, particularly for minority populations [7]. Policies aimed at reducing socioeconomic inequality, improving education and employment opportunities, and ensuring access to affordable housing and healthy food are necessary complements to healthcare policy. These interventions are critical for addressing the root causes of health disparities and improving overall health outcomes for racial and ethnic minorities [6].

### 8.2. Future Research Directions

While the ACA has provided valuable insights into the effects of healthcare reform on minority populations, several areas remain under-researched, offering important opportunities for future investigation. One such area is the long-term impact of Medicaid expansion on health outcomes. While research has shown improvements in healthcare access and utilization, less is known about the sustained effects of the ACA on long-term health outcomes, such as chronic disease management, mental health, and mortality rates [1,2]. Future studies should focus on longitudinal analyses to better understand how continuous insurance coverage affects health outcomes over time, particularly for conditions that disproportionately affect minority populations, such as diabetes, heart disease, and cancer [4].

Additionally, more research is needed on the role of non-expansion states in perpetuating healthcare disparities. While there is substantial evidence that Medicaid expansion has improved healthcare access and outcomes in expansion states, less is known about the specific barriers that prevent non-expansion states from closing the coverage gap. Future research should explore the political, economic, and social factors that have contributed to the resistance to Medicaid expansion and propose targeted interventions to overcome these barriers [5]. This research could help inform future policy decisions aimed at reducing disparities in non-expansion states.

The role of social determinants of health in shaping healthcare utilization and outcomes also warrants further investigation. While the ACA has expanded healthcare access, many minority populations continue to face significant barriers related to poverty, education, and housing, which limit their ability to fully benefit from health insurance coverage [6]. Future research should explore how policies that address social determinants of health can complement healthcare reforms and improve health outcomes for minority populations [7]. This could include studies on the impact of housing stability, access to healthy food, and employment opportunities on healthcare utilization and health outcomes.

Finally, the COVID-19 pandemic has highlighted the potential for telemedicine to reduce healthcare disparities, particularly in rural and underserved urban areas [9]. Future research should examine the effectiveness of telemedicine in improving healthcare access for racial and ethnic minorities and evaluate the barriers to its adoption, such as lack of access to technology and broadband Internet. Telemedicine offers a promising solution to reduce geographic barriers to care, but more research is needed to understand its potential role in reducing healthcare disparities [4].

## 9. Conclusions

The ACA represents a significant step forward in addressing healthcare disparities and expanding access to health insurance for millions of Americans, particularly racial and ethnic minorities. Medicaid expansion and the elimination of cost-sharing for preventive services have played a central role in reducing uninsured rates and increasing healthcare utilization among minority populations [1,3]. However, the uneven adoption of Medicaid expansion, particularly in southern and non-expansion states, has left many racial and ethnic minorities without access to affordable healthcare, perpetuating disparities in health outcomes [5,7]. While the ACA has made important strides in improving healthcare access, significant challenges remain, particularly in non-expansion states where uninsured rates for Black and Hispanic/Latinx adults remain disproportionately high [4,6]. Additionally, affordability continues to be a barrier to care for many low-income individuals, and cultural and linguistic barriers further limit access to quality healthcare for non-English speaking populations [9,16].

Addressing these challenges will require targeted policy interventions, including the nationwide expansion of Medicaid, enhanced affordability measures, and investments in culturally competent care [8,19].

Future research should focus on understanding the long-term impact of the ACA on health outcomes for minority populations, as well as the role of non-expansion states in perpetuating healthcare disparities [2,5]. Additionally, research on the intersection of healthcare policy and social determinants of health is critical for informing future reforms aimed at reducing disparities and improving health outcomes for racial and ethnic minorities [6,7]. By addressing these challenges, policymakers can build on the progress made by the ACA and work toward achieving true health equity for all Americans. A key limitation of this review is that it does not meet the full methodological rigor of a formal systematic review. Risk-of-bias assessments, formal quality scoring, and quantitative synthesis (e.g., meta-analysis) were not conducted. As such, findings should be interpreted in the context of a structured narrative review rather than a systematic evidence appraisal. Future work may build upon this review using systematic methodologies to assess the quality and robustness of the included studies.

To contextualize findings within a broader policy evaluation framework, this review draws on the literature on policy design and capacity. As Howlett (2011) and Wu, Ramesh, and Howlett (2015) argue, policy outcomes depend not only on the intended mechanisms but also on the capacities of the institutions, actors, and tools used in policy delivery [24,25]. The Affordable Care Act embodies a complex blend of design instruments—such as mandates, subsidies, and Medicaid expansion—that require multi-level implementation and coordination. Furthermore, Schneider and Ingram’s (1990) framework on social construction and target populations underscores how public perceptions and political narratives influence both policy design and differential uptake among marginalized communities [23].

By integrating these perspectives, the review interprets racial and ethnic disparities in ACA outcomes as a function of both structural inequities and policy design challenges. Future efforts to advance health equity must consider not only insurance expansion but also the institutional, geographic, and cultural factors that condition healthcare access. A more intersectional and context-sensitive policy framework is essential to ensure that the gains from reforms like the ACA are equitably shared across all racial and ethnic communities.

## Figures and Tables

**Figure 1 ijerph-22-01059-f001:**
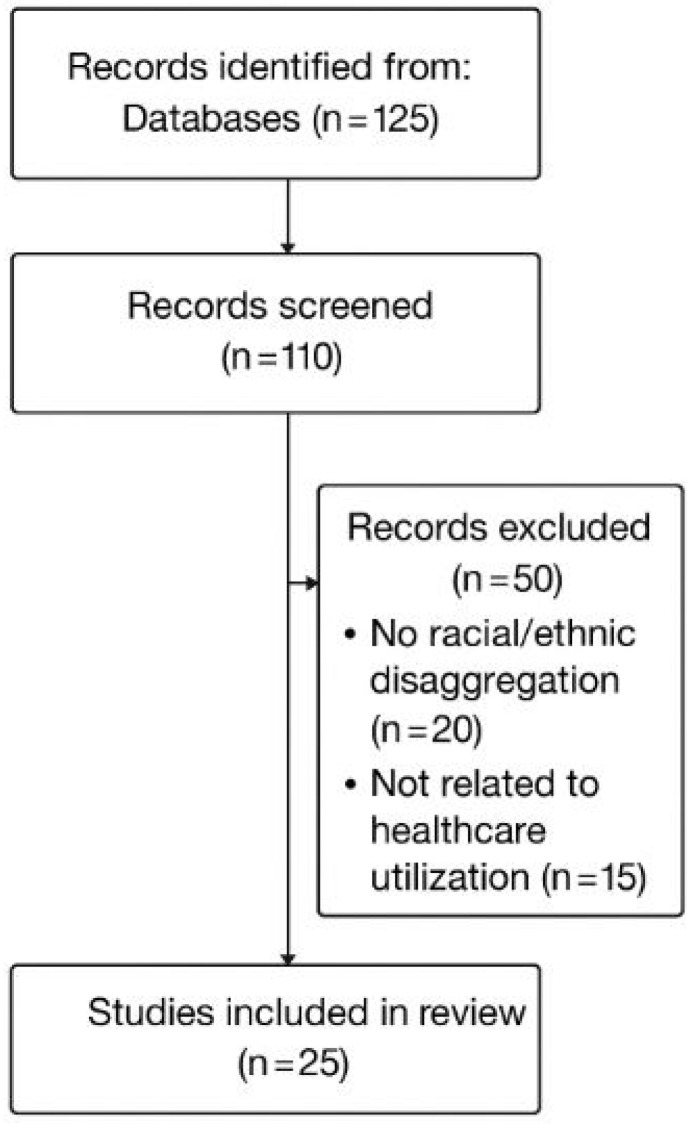
PRISMA flow diagram for study selection process.

**Figure 2 ijerph-22-01059-f002:**
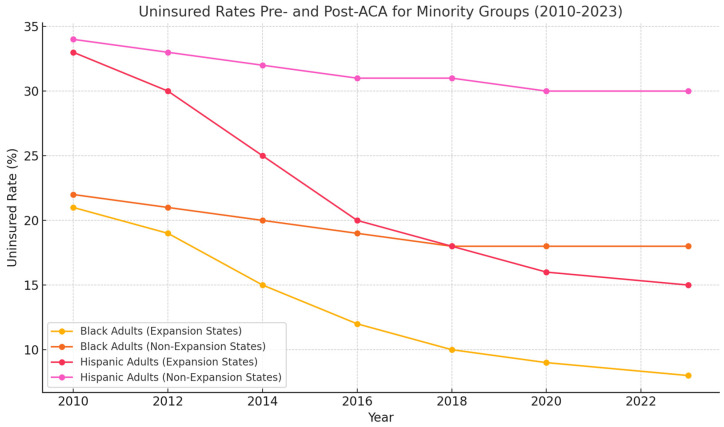
Uninsured rates for Black and Hispanic/Latinx adults in Medicaid expansion and non-Expansion States, 2010–2023.

**Figure 3 ijerph-22-01059-f003:**
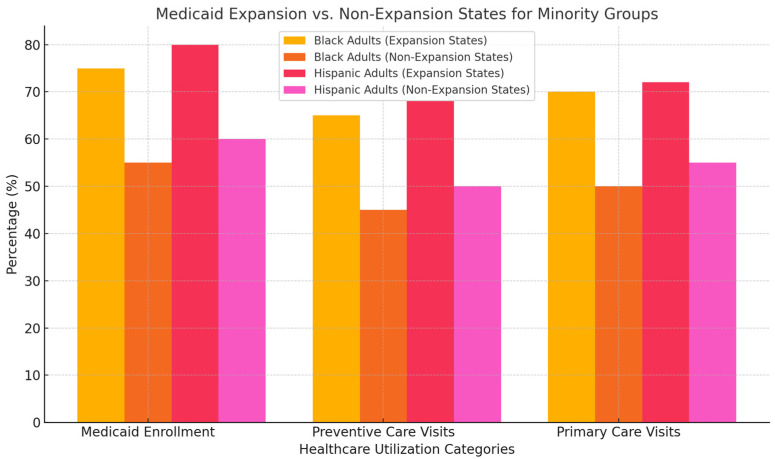
Comparison of Medicaid enrollment, preventive care visits, and primary care visits among Black and Hispanic/Latinx adults in expansion vs. non-expansion states.

**Figure 4 ijerph-22-01059-f004:**
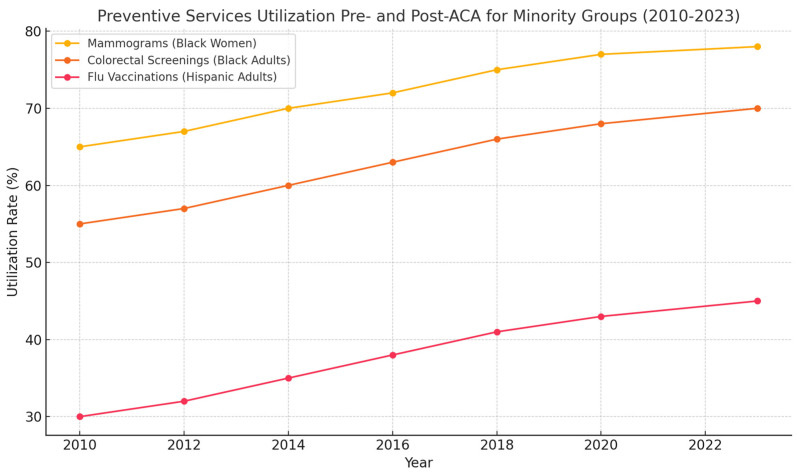
Increased utilization of preventive services among Black and Hispanic/Latinx adults, 2010–2023.

**Figure 5 ijerph-22-01059-f005:**
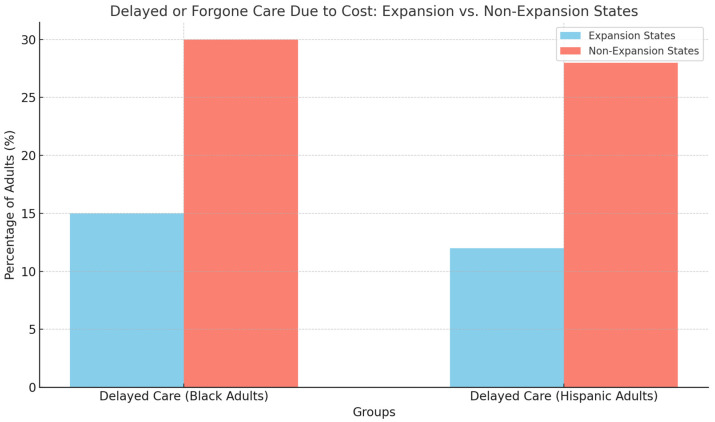
Percentage of Black and Hispanic/Latinx adults delaying or forgoing care due to cost in expansion vs. non-Expansion states.

## Data Availability

This study is a narrative literature review and does not involve the generation or analysis of primary or secondary quantitative datasets. All sources used in the review are publicly available and cited within the manuscript. Readers can access the referenced peer-reviewed articles, government reports, and policy documents through academic databases such as PubMed, JSTOR, and Google Scholar.

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
