# Peer review of "Unequal Gains? A Literature Review on the Affordable Care Act’s Effects on Healthcare Utilization Across Racial and Ethnic Groups"

_ijerph, 2025, doi:10.3390/ijerph22071059_

Round 1

Reviewer 1 Report

Comments and Suggestions for Authors

This article provides a comprehensive literature review of the Affordable Care Act's (ACA) impact on healthcare utilization, with particular attention to racial and ethnic disparities. The study focuses on insurance coverage, preventive services, and health outcomes among minority populations.

1. The introduction presents a relevant and timely topic and frames the ACA within the broader context of racial disparities in healthcare.

-  The objectives of the paper are clear, but the research gap could be better emphasized, for example what is missing from existing reviews? What novel perspective does this one bring?---

2. The review of ACA provisions is well-organized and thorough, covering Medicaid expansion, insurance marketplaces, preventive services, and provisions for cultural competence. But the discussion is largely descriptive and would benefit from more critical synthesis or comparative analysis across sources.

3. Regarding the methodology, the selection of 52 peer-reviewed articles and policy reports, with a preference for quasi-experimental methods (DiD, IV, RD), is appropriate and clearly stated.

- The methodology for literature inclusion is a strong point, offering transparency about scope and rigor and the division into empirical studies and policy reports is also effective in providing both evidence and context.

4. The discussion part is generally well formulated, but it often repeats findings from referenced studies without integrating them or proposing theoretical explanations. There is also limited attention to intersectionality (e.g., race × geography × gender).

5. The conclusion successfully summarizes the main findings and reiterates the importance of equitable healthcare policy.

6. My English knowledge is not sufficient to comment on the quality of the writing, while generally I did not find any issues with it.

Reviewer 2 Report

Comments and Suggestions for Authors

The study presents a very nice and professional review of literature on the topic. It is thoroughly described how the literature search was performed and which topics prioritized. I have only minor and merely of a check to be safe nature:

  1. Whole the study is clear on what was included in the review, it should pay some attention also to what was not included, i.e. which topics were excluded. Not all of course, but the essential ones should be mentioned.
  2. In extension hereof, it would be nice with a brief discussion of these topics and potential implications of their absence.
  3. finally, I notice that the number of references is fairly limited (22). The authors should briefly discuss and/or reflect on whether this figure is sufficient.

I wish you good luck with the revision and publication of a good paper.

Reviewer 3 Report

Comments and Suggestions for Authors

The article addresses a relevant and current topic. However, it lacks methodological rigor and clear theoretical grounding.

Regarding methodological rigor, it is essential that the methods of data collection and analysis be described in sufficient detail to allow for replication. In this respect, I recommend that the authors use the PRISMA statement as a reference for reporting the review (Page, Matthew J. et al. “The PRISMA 2020 statement: an updated guideline for reporting systematic reviews.” BMJ vol. 372, n71, 29 Mar. 2021. doi:10.1136/bmj.n71).

As it stands, several questions remain unanswered, such as:

  • How were the 52 studies selected?

  • What search queries were used?

  • What were the inclusion and exclusion criteria?

  • Why were PubMed, JSTOR, and Google Scholar chosen? Why were more comprehensive databases such as Scopus and Web of Science not used?

As for theoretical grounding, since the authors aim to evaluate the effectiveness of a public policy, it would be important to incorporate relevant literature in this field, such as policy design, policy capacity, or policy analysis. This literature would support a broader and more in-depth discussion of the factors contributing to the success or failure of the policy. This would also help guide the interpretation of the studies included in the review.

Round 2

Reviewer 3 Report

Comments and Suggestions for Authors

The author adequately addresses all my recommendations, and resolves the questions that, from my perspective, were not answered initially. In fact, the quality of the manuscript improved considerably after the revision. In this sense, I see no further obstacle to publication.